# The MAPK substrate MASS proteins regulate stomatal development in Arabidopsis

Xueyi Xue[1]☉¤, Chao Bian[1,2]☉, Xiaoyu Guo[1], Rong Di[2], Juan Dong[1,2]*

**1** The Waksman Institute of Microbiology, Rutgers, the State University of New Jersey; Piscataway, New Jersey, United States of America, **2** Department of Plant Biology, Rutgers, the State University of New Jersey, New Brunswick, New Jersey, United States of America

☉ These authors contributed equally to this work.
¤ Current address: Department of Plant Biology, University of Illinois at Urbana-Champaign, Urbana, Illinois, United States of America
* dong@waksman.rutgers.edu

**Data Availability Statement:** All relevant data are within the manuscript and its Supporting Information files.

## Abstract

Stomata are specialized pores in the epidermis of the aerial parts of a plant, where stomatal guard cells close and open to regulate gas exchange with the atmosphere and restrict excessive water vapor from the plant. The production and patterning of the stomatal lineage cells in higher plants are influenced by the activities of the widely-used mitogen-activated protein kinase (MAPK) signaling components. The phenotype caused by the loss-of-function mutations suggested pivotal roles of the canonical MAPK pathway in the suppression of stomatal formation and regulation of stomatal patterning in Arabidopsis, whilst the cell type-specific manipulation of individual MAPK components revealed the existence of a positive impact on stomatal production. Among a large number of putative MAPK substrates in plants, the nuclear transcription factors SPEECHLESS (SPCH) and SCREAM (SCRM) are targets of MAPK 3 and 6 (MPK3/6) in the inhibition of stomatal formation. The polarity protein BREAKING OF ASYMMETRY IN THE STOMATAL LINEAGE (BASL) is phosphorylated by MPK3/6 for localization and function in driving divisional asymmetries. Here, by functionally characterizing three MAPK SUBSTRATES IN THE STOMATAL LINEAGE (MASS) proteins, we establish that they are plasma membrane-associated, positive regulators of stomatal production. MPK6 can phosphorylate the MASS proteins *in vitro* and mutating the putative substrate sites interferes the subcellular partition and function of MASS in planta. Our fine-scale domain analyses identify critical subdomains of MASS2 required for specific subcellular localization and biological function, respectively. Furthermore, our data indicate that the MASS proteins may directly interact with the MAPKK Kinase YODA (YDA) at the plasma membrane. Thus, the deeply conserved MASS proteins are tightly connected with MAPK signaling in Arabidopsis to fine-tune stomatal production and patterning, providing a functional divergence of the YDA-MPK3/6 cascade in the regulation of plant developmental processes.

**Funding:** JD is supported by grants from the National Institute of Health (R01GM109080 and R35GM131827). XX was supported by the Charles and Johanna Busch Fellowship from Rutgers. CB was supported by fellowships from the Chinese Scholar Council and Rutgers University. The funders had no role in study design, data collection and analysis, decision to publish, or preparation of the manuscript.

**Competing interests:** The authors have declared that no competing interests exist.

## Author summary

Stomata surrounded by guard cells are breathing pores in the plant epidermis, where they open to allow gas exchange and close to restrict water loss. The production and patterning of stomata in the model plant Arabidopsis provide an ideal genetic and cell biological system for studying the molecular mechanisms underlying developmental program and plasticity in responding to environmental changes. The MAPK cascades are ubiquitous signaling modules in eukaryotes. They regulate diverse cellular programs by relaying extracellular signals to intracellular regulators. In the model plant Arabidopsis, MAPK 3 and 6 were found to phosphorylate several protein substrates in the nucleus and cytoplasm to regulate stomatal development and patterning. In this study, we report that a group of new MAPK substrates, the MASS proteins, function at the plasma membrane to regulate stomatal production and patterning in Arabidopsis. Thus, the output of MAPK signaling in the regulation of stomatal development is diverged by differentially localized substrates, suggesting that the concerted activities of MAPK substrates fine-tune stomatal development to ultimately improve plant adaptability to the changing environment.

## Introduction

The mitogen-activated protein kinase (MAPK) cascades are central signaling pathways that regulate a wide range of cellular processes in plant growth, development and stress responses [1–3]. They function downstream of the cell-surface receptors to deliver and amplify extracellular stimuli that trigger a myriad of cytoplasmic and nuclear responses [4]. Stomatal development and patterning in the model plant Arabidopsis are tightly regulated by a canonical MAPK signaling cascade composed of the MAPKK kinase YODA (YDA), MAPK Kinase 4 and 5 (MKK4/5) and MAPK 3 and 6 (MPK3/6) [5–7]. Mutants and genetic analyses established a pivotal role of this YDA MAPK signaling pathway in suppressing Arabidopsis stomatal production at early developmental stages [5–7]. On the other hand, at later developmental stages, a positive regulation on stomatal proliferation was identified that seemed to be achieved by a differently assembled MAPK module of YDA-MKK7/9-MPK3/6 and other unknown MAPKs [7, 8].

The MAPK cascades control a diverse variety of biological processes that are achieved by the regulation of a plethora of substrates. In stomatal development, several key factors are modified and regulated by MAPKs. The stomatal lineage initiation is controlled by the bHLH transcription factors SPEECHLESS (SPCH) [9, 10] and its partners SCREAM/ICE1 (SCRM/ICE1) and SCRM2 [11]. MPK3/6 phosphorylate SPCH for protein degradation, thus providing a mechanistic link to the suppression of stomatal production [12]. SCRM/ICE1 is also phosphorylated by MPK3/6, so that SCRM/ICE1 protein stability was reduced in cold tolerance [13]. Additionally, recent studies showed that SCRM/ICE1 physically bridges MAPKs and SPCH to initiate the stomata lineage [14].

Besides these strong negative regulation of MPK3/6 signaling in the early stages of stomatal development [6, 12], a positive role of the YDA-MKK7/9-MPK3/6 at the late stages was also suggested by the stage-specific manipulation of different tiers of this MAPK cascade in Arabidopsis [7, 8]. However, what substrate/s control this cell fate flip remains unknown. Previously, a few collections of putative MAPK substrate proteins were predicted by peptide library screening combined with bioinformatics analysis [15], protein-protein interaction based on yeast two-hybrid screening [16, 17], and *in vivo* phosphoproteomic studies [18]. However,

detailed functional characterization of these proteins requires significant endeavor, thus the predicted candidates have been seldom further pursued.

The YDA-MKK4/5-MPK3/6 MAPK signaling pathway functions downstream of the plasma membrane receptor-like proteins (RLPs) and kinases (RLKs), including TOO MANY MOUTHS (TMM), the ERECTA family and the Somatic Embryogenesis Receptor Kinase (SERK) family [19–23]. Upstream of the MAPKKK YDA, a few regulators have been characterized for their functions in plant development, including the SHORT SUSPENSOR (SSP) receptor-associated kinase [24] and the G protein subunit Gβ in zygotic development [25], the GSK3-like BRASSINOSTEROID INSENSITIVE 2 (BIN2) kinase [26, 27] and a MAPK scaffold polarity protein BREAKING OF ASYMMETRY IN THE STOMATAL LINEAGE (BASL) in stomatal asymmetric cell division [28]. Their modulation of YDA's function can be achieved by enzymatic inhibition/activation, physical scaffolding for signal specificity and spatiotemporal restriction, *etc*.

Previously, Sörensson *et al.* [15] determined consensus phosphorylation sequences for MPK3/6 in Arabidopsis. They found one of the substrates At1g80180 was phosphorylated by MAPKs and overexpression of it induced stomatal overproduction and clustering [15]. Here, we performed in-depth functional genetics to characterize the three MAPK SUBSTRATES IN THE STOMATAL LINEAGE (MASS) proteins that At1g80180 belongs to for their biological functions in Arabidopsis stomatal development. We found that the MASS proteins are associated with the plasma membrane where they promote the stomatal formation and regulate stomatal patterning. We provide experimental evidence supporting that MAPKs can phosphorylate the MASS proteins *in vitro* and the putative substrates sites may regulate the MASS subcellular localization and protein function, and in turn the MASS proteins interact with YDA at the plasma membrane, possibly suppressing YDA function. Thus, the functional connection between the MASS family and the YDA MAPK cascade provides a new angle to study how external signals through MAPKs fine-tune stomatal development at the plasma membrane.

## Results

### MASS proteins positively regulate stomata formation

The previous work by Sörensson et al. showed that At1g80180 is a substrate of MPK3 and MPK6 and overexpression of a phosphor-mimicking version of the protein seemed to generate stomatal overproduction and clustering [15]. Inspired by that, we investigated its biological function in stomatal development and possible functional interaction with the core YDA MPK3/6 pathway in Arabidopsis. At1g80180 encodes a short protein (15 kD) with unknown functions and belongs to a small family of three in the Arabidopsis genome (At1g15400 and At5g20100) (Fig 1A). We overexpressed the three genes either in the stomatal lineage cells by using a cell type-specific *BASL* promoter or ubiquitously by using the CaMV *35S* promoter. The results showed that all the transgenic populations produced similar stomatal phenotypes: overproliferated stomatal guard cells in a clustered pattern (Fig 1B–1D and S1A Fig). The elevated transcript levels in these overexpression lines were demonstrated by real-time PCR (Fig 1E). The phenotypes suggest that all three genes might promote stomatal production and regulate stomatal patterning, therefore they are named as *MAPK SUBSTRATES IN THE STOMATAL LINEAGE* (*MASS*) *1*, *2*, and *3* (Fig 1A).

To functionally characterize the three *MASS* genes, we analyzed their promoter activity by examination of the transcriptional reporter lines that drive the expression of nuclear YFP (nucYFP) in plants. We found that all three promoters were broadly active in the leaf epidermal cells albeit with preferred cell type-specificity (S1B Fig) and in some other tissues, e.g. the

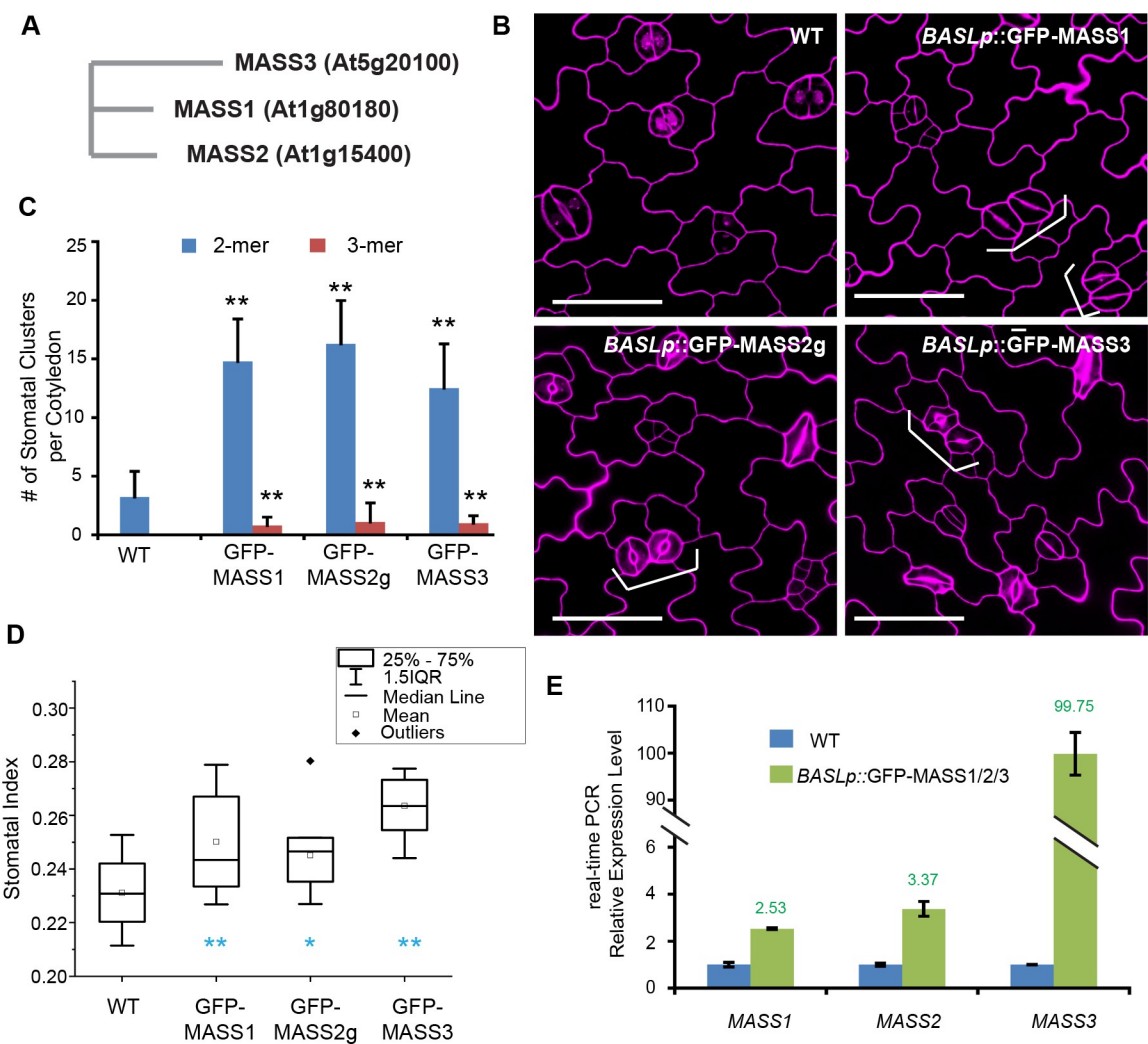

**Fig 1. Stomatal phenotypes caused by MASS overexpression.** (A) Phylogenetic tree of the Arabidopsis MASS family. (B) Stomatal phenotypes induced by MASS1/2/3. Confocal images of 3-dpg adaxial side of cotyledon epidermis in WT (Col) and overexpression seedlings of GFP-MASS1, GFP-MASS2g, and GFP-MASS3, all driven by the *BASL* promoter. Cell outlines were stained with propidium iodide (PI, magenta). Brackets indicate stomatal clusters. Scale bar represents 50 μm. (C) Quantification of the numbers of stomatal clusters per cotyledon in 10-dpg seedlings of the designated plants. (D) Quantification of stomatal index (SI) in 5-dpg adaxial cotyledons of the designated seedlings. (E) Quantitative real-time PCR analysis of *MASS1/2/3* expressions in WT and GFP-MASS transgenic lines in (B). *significantly different compared with the WT (Col) values (Student's *t*-test, *P < 0.05, **P < 0.01).

hypocotyl and root, at the seedling stage (S1B Fig). We collected single T-DNA insertional mutants and crossed them to generate double and triple mutants (Fig 2A and S2A Fig). The transcript levels of the three genes in the triple *mass1;2;3* mutants were assessed by real-time PCR (Fig 2E) and the data show that, while *mass3* is a knock-down, *mass1* and *mass2* are null mutations. Quantification of stomatal production in 5-day old cotyledons suggested that the triple mutant produced lowered stomatal index (Fig 2A and 2F), consistent with the overexpression data (Fig 1B–1D and S1A Fig), supporting a positive role of the *MASS* genes in stomatal production. We also characterized the lower-order mutants (singles and doubles) and found that the double mutant *mass 1;3*, but not *mass1;2* and *mass2;3*, produced reduced numbers of stomata, though to a less extent when compared with those of the triple mutants (Fig 2F). None of the three single mutants showed any discernable defects in general growth and

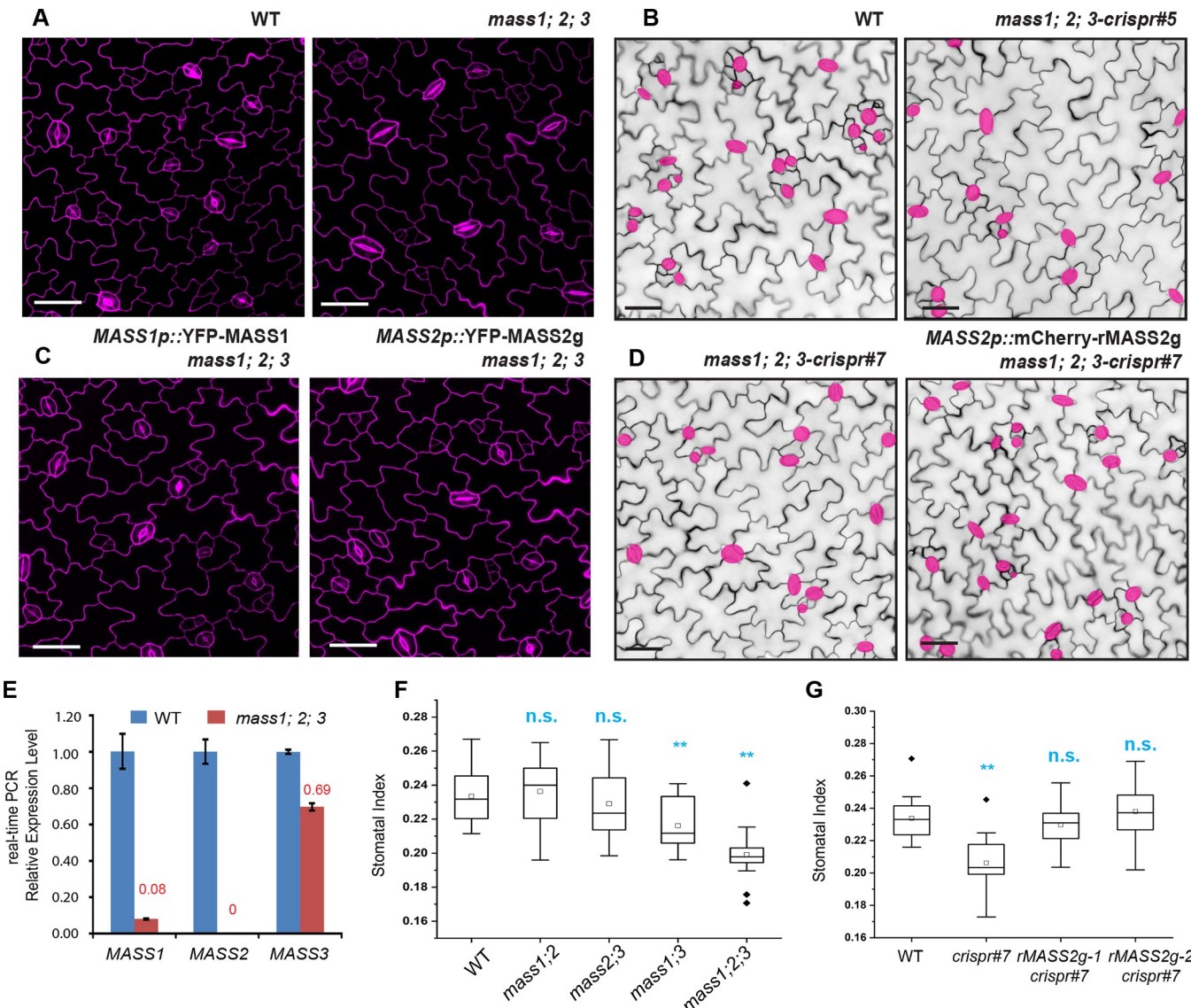

**Fig 2. MASS positively regulates stomata formation.** (A) Confocal images of 3-dpg adaxial side of the cotyledon epidermis in WT (Col) and *mass1;2;3* mutant seedlings. (B) DIC images of 5-dpg cotyledons in WT and *mass1;2;3-crispr #5* seedlings. (C) Confocal images of 3-dpg adaxial side of the cotyledon epidermis in complementation lines *MASS1p*::YFP-MASS1 and *MASS2p*::YFP-MASS2g in *mass1;2;3* background. (D) DIC images of 5-dpg cotyledons in *mass1;2;3-crispr #7* and complementation seedlings expressing the crispr-resistant version of MASS2g (rMASS2g). Cell outlines in (A-D) were stained with PI (magenta). Guard cells in (B, D) were highlighted in pink for better visualization. Scale bars represent 50 µm in (A-D). (E) Quantitative real-time PCR analysis of *MASS1/2/3* expressions in WT and *mass1;2;3* mutant. (F) Quantification of SI in 5-dpg adaxial cotyledons of T-DNA mutants. (G) Quantification of SI in 5-dpg adaxial cotyledons of *mass1;2;3-crispr #7* mutant and the designated complementation lines. *significantly different compared with the WT (Col) values (Student's *t*-test, *P < 0.05, **P < 0.01). n.s: not significant.

stomatal development. Thus, our mutant analyses indicated a redundantly positive role of the three *MASS* genes in stomatal production.

To consolidate the loss-of-function phenotypes, we deployed the CRISPR/Cas9-mediated genome-editing strategy [29] to create genetic lesions in all three *MASS* genes (sgRNA positions shown in S2A Fig). The wild-type plants Columbia-0 (Col) plants were transformed with the CRISPR/Cas9 construct that carried three sgRNAs, each of which should specifically target one of the three *MASS* genes. Two independent triple mutant lines (T3) were established

(*mass1;2;3-crispr#5* and *#7*, see the mutated sequences and genotyping data in S2B and S2C Fig), in which three genes were either early terminated in translation or made internal deletions (S2C Fig). Phenotypic characterization of stomatal development suggested that the CRISPR-generated mutants were similar to the T-DNA mutagenized triple mutants (Fig 2B and 2D, and S2D Fig), in both of which stomatal indices were lowered to around 20% compared to that of the wild-type (23%, n = 14 individual plants) (Fig 2F and 2G).

With respective to genetic complementation, we introduced the expression of N-terminal YFP fused MASS1 and MASS2, both driven by their endogenous promoters, into the T-DNA *mass1;2;3* mutants (Fig 2C, S3A and S3B Fig). We also introduced a CRISPR/Cas9-resistant version of mCherry-rMASS2 (rMASS2 containing nucleotide mutations in the sgRNA targeting site without changing the MASS2 amino acid sequence) into *mass1;2;3-crispr#7* plants (Fig 2D). In both cases, we found that the mutant phenotypes were recovered by YFP-MASS1/2 (Fig 2C and 2D) and the lowered stomatal indices in the *crispr* mutants were recovered by expression of mCherry-rMASS2 back to the wild-type levels (Fig 2G). Again, both MASS1 and MASS2 protein expressions were detected in the leaf tissues, with more abundant expression of *MASS2* in the epidermis and *MASS1* in the mesophyll layer (S3A and S3B Fig). The subcellular distribution patterns of MASS1 and MASS2 (S3A and S3B Fig) in *Arabidopsis* were comparable with those in tobacco epidermal cells (Fig 3A and 3B); both were localized to the nucleus and the plasma membrane (see below for more details). In parallel, we also generated C-terminal tagged MASS proteins for complementation. Because the orientation of such fusions may disturb the C-terminal motif that is critical for MASS function (see below about MASS2 subdomains), they were therefore abandoned for further analysis. Thus, collectively, our genetic evidence (loss-of-function and overexpression) suggested that the three MASS genes may redundantly contribute to promoting stomatal production in Arabidopsis.

## MASS functions at the plasma membrane

By amino acid sequence analysis, no functionally annotated domains can be recognized in the three MASS proteins. To characterize the biological functions of these novel regulators, we analyzed their protein subcellular localization by examining the N-terminal fluorescent protein-tagged MASS proteins in both tobacco epidermal cells and Arabidopsis stomatal lineage cells (Fig 3A–3F). The genomic regions of *MASS1* and *MASS3* do not contain introns, thus their genomic/coding sequences were amplified for constructing the reporter lines (CFP/YFP-MASS1/3). As three alternative splicing sites were annotated for *MASS2* (S2A Fig), we amplified the genomic region flanking all three variants to generate the reporter line (CFP/YFP-MASS2g). The localization data in tobacco cells showed that the three MASS proteins were differentially distributed at the subcellular level: MASS1 and MASS2 appeared in the nucleus and at the plasma membrane, whilst MASS3 predominantly localized at the plasma membrane (Fig 3A–3C). When the three genes were expressed in the stomatal lineage cells (driven by the *BASL* promoter), they showed consistently differential expression patterns as in tobacco epidermal cells, with MASS1 and MASS2 dually localized in the nucleus and at the plasma membrane, while MASS3 mainly at the plasma membrane (Fig 3D–3F).

Because the MASS2 protein showed expression at both subcellular locations and overexpression of it generated elevated number and clustered stomatal lineage cells (Fig 3B, 3E and 3G), we tested where its biological location is, in the nucleus, or at the plasma membrane, or both. As in the MASS protein sequences, no signal peptides for apoplast secretion or transmembrane domains to span the membranes were predicted, we added a myristoylation lipid modification site [30] to artificially tether MASS2 to the plasma membrane. Indeed, myr-GFP-MASS2g was found exclusively at the plasma membrane and, interestingly, overexpression of this membrane-

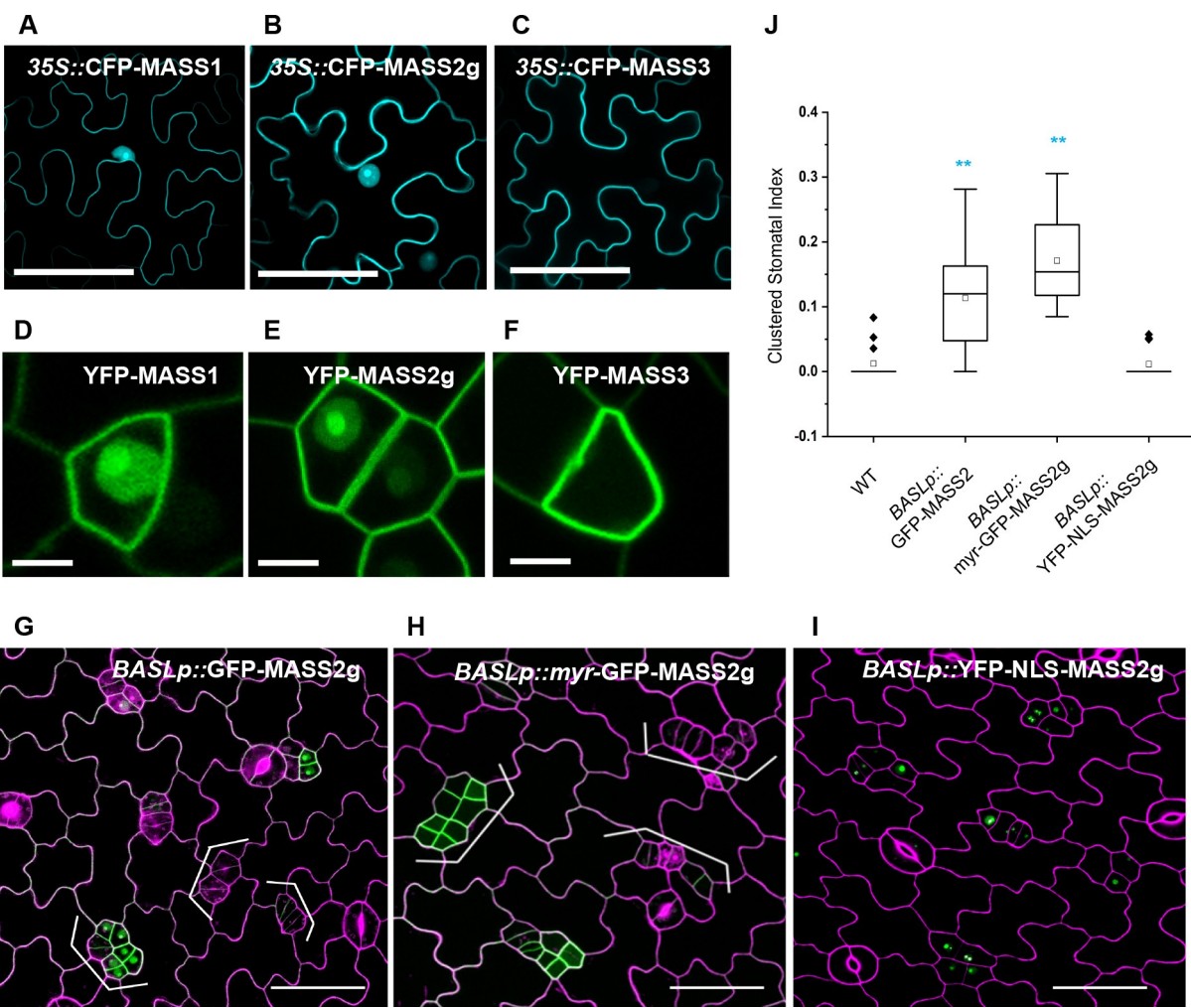

**Fig 3. MASS functions at the plasma membrane.** (A-C) Protein subcellular localization (cyan) in tobacco epidermal cells leaves. Confocal images of tobacco epidermis transiently expressing CFP-MASS1 (A), CFP-MASS2g (B), and CFP-MASS3 (C), all of which were driven by the *35S* promoter. Scale bars represent 50 μm. (D-F) Protein localization (green) in the stomatal lineage cells in Arabidopsis. Confocal images of 3-dpg adaxial cotyledon epidermis expressing YFP-MASS1 (D), YFP-MASS2g (E), and YFP-MASS3 (F), all driven by the *BASL* promoter. Scale bars represent 10 μm. (G-I) Stomatal phenotypes triggered by expressing MASS2 and other variants. Confocal images of 3-dpg adaxial cotyledon epidermis expressing GFP-MASS2g (G), myr-GFP-MASS2g (H), and YFP-NLS-MASS2g (I). Green shows GFP-MASS expression pattern. Scale bars represent 50 μm. Brackets indicate stomatal clusters and abnormal cell divisions. (J) Index of clustered stomata in 5-dpg adaxial cotyledons of designated plants. Cell outlines in (G-I) were stained with PI (magenta). * significantly different compared the WT (Col) (Student's *t*-test, *P < 0.05, **P < 0.01).

attached version recapitulated, and even slightly enhanced, the stomatal clustering phenotypes caused by the overexpression of the wild-type protein (Fig 3H and 3J). On the other hand, when fused with a nuclear localization signal (NLS), YFP-NLS-MASS2g showed the anticipated nuclear-only pattern, but this version did not induce any obvious stomatal phenotypes (Fig 3I and 3J). The same strategies were applied to the dually localized MASS1 protein as well, and the data consistently show that plasma membrane-only but not nuclear-localized MASS1 promoted stomatal production (S3C and S3D Fig). Thus, collectively, our data suggested that the positive regulation of the MASS proteins in stomatal development arises from the plasma membrane pool, but not from the nuclear pool.

## Fine-scale analysis of the MASS2 subdomains

As the MASS2 proteins do not contain transmembrane domains, its association with the plasma membrane might be achieved by protein-protein or protein-lipid interactions in plant cells. Considering MASS2 has three splicing variants (S4A Fig), we first examined their subcellular localization (N-terminal GFP fusions driven by the *BASL* promoter) and found that all of them showed the typical dual localization with slight differences in the preferential partition (S4B Fig). MASS2.3, as the longest one (S4A Fig), was used as a representing member to align with MASS1 and 3 for subdomain analysis (S4C Fig). The three MASS proteins show high similarity at the two terminal regions but are not conserved in the middle (Fig 4A and S4C Fig). We thus split MASS2.3 into two halves to make GFP-tagged MASS2.3_N76 and MASS2.3_C72, both driven by the *BASL* promoter (Fig 4A). The results showed that both truncations failed to localize correctly. The first half GFP-MASS2.3_N76 lost the distinct localization in the nucleus, whereas the second half GFP-MASS2.3_C72 lost the plasma membrane localization (Fig 4B–4D). To further narrow down the critical segments for specific localization at the plasma membrane and in the nucleus, respectively, we deleted the highly conserved regions at the two ends to create GFP-MASS2.3_Δ29N (the N-terminal 29 amino acids deleted), GFP-MASS2_Δ13C and GFP-MASS2_Δ25C (the C-terminal 13 and 25 amino acids deleted, respectively) (Fig 4A). The subcellular localization data clearly demonstrated that GFP-MASS2.3_Δ29N was only shown in the nucleus, suggesting the N-terminal 29-aa is required for the plasma membrane-association, while GFP-MASS2_Δ13C was only found at the plasma membrane, supporting that the C-terminal 13-aa determines the nuclear accumulation (Fig 4E–4G). Interestingly, the plasma membrane-only GFP-MASS2_Δ13C promoted stomata production and clustering, a phenotype resembling that of the full-length MASS2 overexpression (Fig 4H and 4I), again supporting the biological function of MASS2 at the plasma membrane. However, the further shortened MASS2.3_Δ25C, albeit successfully localized to the plasma membrane, failed to induce stomatal overproduction (Fig 4J), hinting the critical role of the small region between Δ13C and Δ25C for its biological function in stomatal development (marked in Fig 4A). In addition, none of the other shortened versions, MASS2.3_N76, MASS2.3_C72 and MASS2.3_Δ29N, were sufficient to trigger this phenotype (S4D Fig), probably due to the lack of either plasma membrane association or the critical functional region. Taken together, we established that the MASS proteins promote stomatal formation at the plasma membrane and we defined three specific regions in MASS2.3 that are important for nuclear accumulation, plasma membrane association, and the biological function at the plasma membrane, respectively (Fig 4A).

## MPK6-mediated phosphorylation and possible connection with MASS localization and function

Sörensson et al. (2012) previously demonstrated that MASS1 is phosphorylated by MPK6 *in vitro* kinase assays [15]. We further tested MASS2 recombinant proteins. Because one of the splicing variants, MASS2.2, is most similar to MASS1, we purified the MASS2.2 recombinant proteins and found that MASS2.2 was phosphorylated by *in vitro* constitutively active MKK5 (MKK5$^{DD}$)-activated MPK6 (Fig 5A). In parallel, mutating the serine residue (S107) to alanine (A) in the conserved MAPK-substrate P-P-<u>S</u>-P motif abolished the MPK6-mediated phosphorylation of MASS2.2 (Fig 5A), supporting that MASS2 is phosphorylated by MPK6 at the S107 site *in vitro*, consistent with the previous study that MASS2 was found phosphorylated among isolated plasma membrane-enriched phospho-peptides in Arabidopsis [31].

Phosphorylation may alter protein subcellular localization. We manipulated the phosphorylation site of MASS1 (S105) by generating a phospho-deficient version MASS1$^{S105A}$ and a

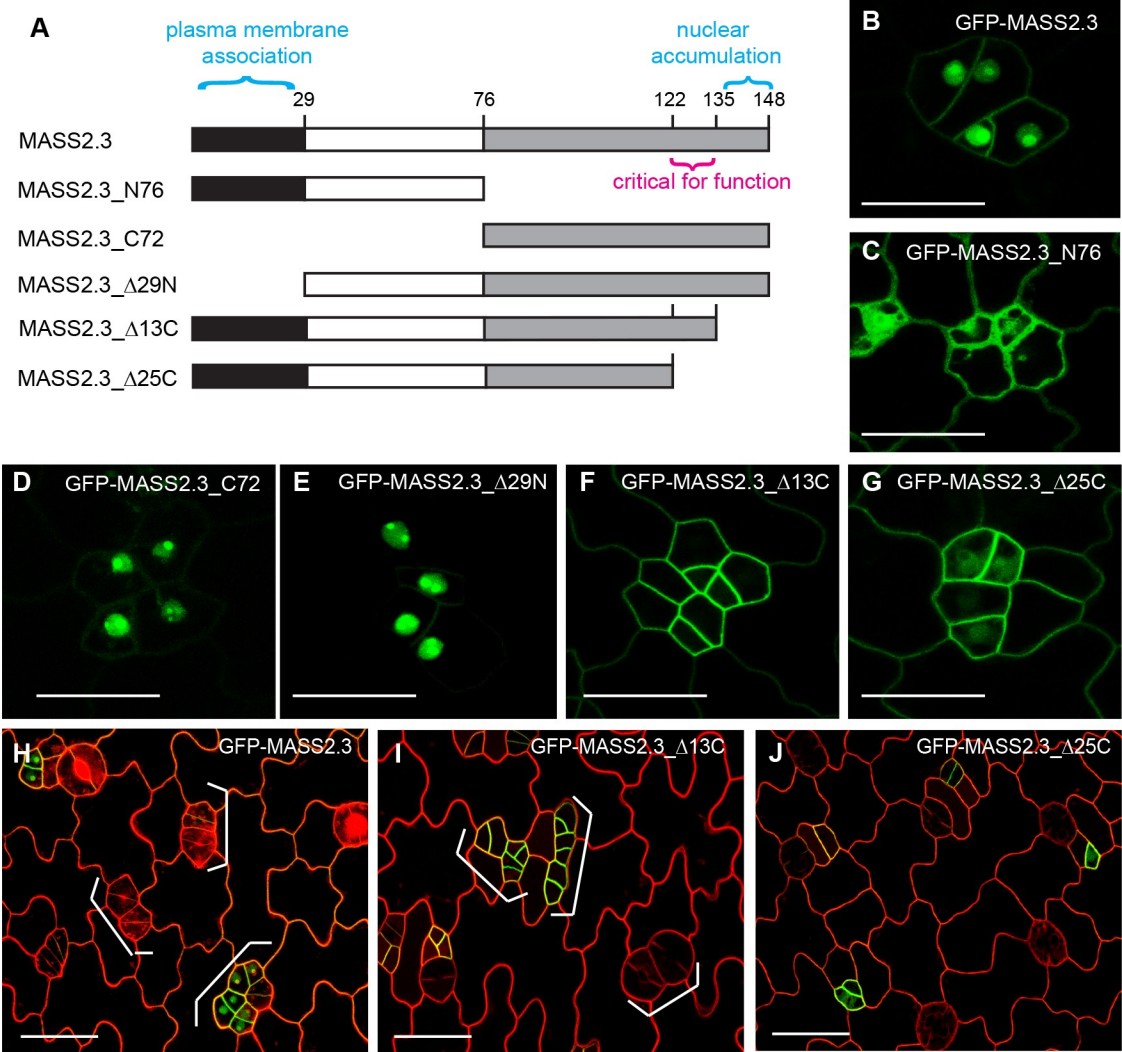

**Fig 4. Fine-scale analysis of the MASS2 subdomains.** (A) Diagram of MASS2.3 subdomains with proposed functions. N76, the N-terminal domain containing 76-aa; C72, the C-terminal domain containing 72-aa; Δ29N, the N-terminal 29-aa deleted; Δ13C, the C-terminal 13-aa delete; Δ25C, the C-terminal 25-aa deleted. (B-G) Confocal images of 3-dpg adaxial cotyledon epidermis showing the localization of GFP-fused MASS2.3 variants (green). (H-J) Confocal images of 3-dpg adaxial cotyledon epidermis showing stomatal phenotypes of expressing GFP-MASS2.3 full-length and truncated versions (green). Brackets indicate stomatal clusters and abnormal cell divisions. Cell outlines in (H-J) were stained with PI. Scale bar represents 20 μm in (B-G) and 50 μm in (H-J).

phospho-mimicking version MASS1$^{S105D}$, respectively. By examining the YFP-tagged proteins, we found that phosphorylation status is influential to MASS1 subcellular distribution because neither MASS1$^{S105A}$ nor MASS1$^{S105D}$ showed robust plasma membrane-association, but both more abundantly accumulated in the nucleus (Fig 5B and 5C and S5A Fig). With respective to the MASS2 localization, because we established that MASS2.2 was phosphorylated by MPK6 *in vitro* (Fig 5A), we assessed the localization pattern of YFP-MASS2.2$^{S107A}$ and YFP-MASS2.2$^{S107D}$ in the stomatal lineage cells. Consistently, the fluorescence intensity profiling results demonstrated that both versions showed reduced abundance at the plasma membrane but elevated accumulation in the nucleus (Fig 5D and S5B Fig). However, we did not detect an obvious change of the PM localization of MASS1 and MASS2 when MPK3/6 signaling was disturbed by overexpression of the dominant-negative *MPK6AEF* or chemically

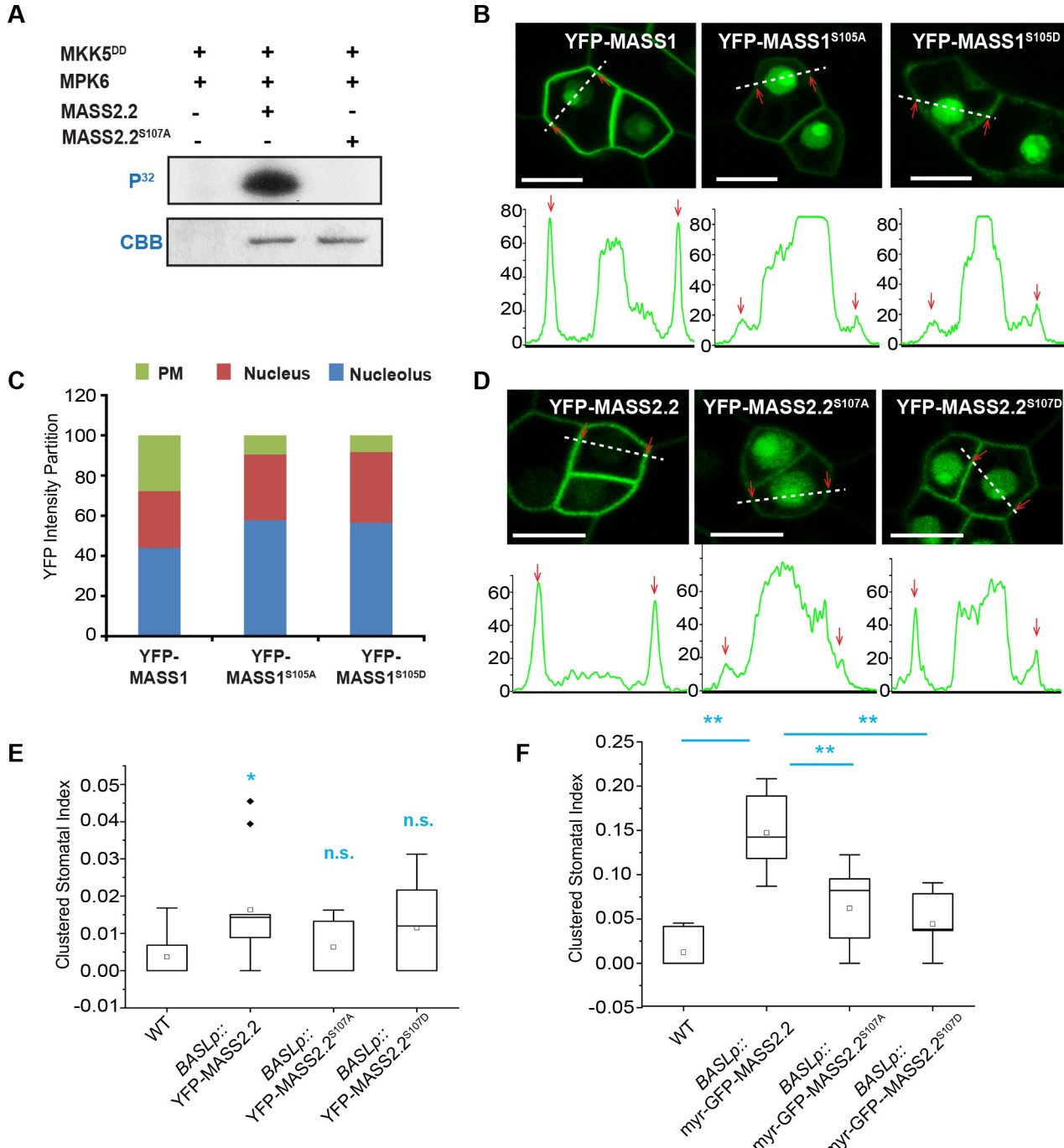

**Fig 5. MPK6-mediated phosphorylation regulates MASS localization and function.** (A) *In vitro* kinase assay showing MKK5$^{DD}$-activated MPK6 phosphorylation of MASS2.2. (B) Upper panel showing the localization of YFP-tagged MASS1 and MASS1 phosphor-variants (green). Lower panel shows the YFP intensity profiling along the lines drawn in the above images. Red arrows indicate YFP signals at the plasma membrane. Scale bars represent 10 μm. (C) Quantification of YFP intensity partition in designated subcellular regions shown in (B). (D) Upper panel, confocal images showing localization of YFP-tagged MASS2.2 and phosphor-variants. Lower panel, intensity profiling of the corresponding YFP signals along the lines drawn in the above images. Red arrows indicate YFP signals at the plasma membrane. Scale bars represent 10 μm. (E, F) Quantification of clustered stomata index in 5-dpg adaxial cotyledons expressing YFP-tagged MASS2.2 and phosphor-variants (E) and myristoylated GFP-tagged proteins as designated (F). * in (E), significantly different compared with the WT (Col) values. ** in (F) significantly different between the two samples being compared (bars). (Student's *t*-test, *P < 0.05, **P < 0.01). n.s: not significant.

inhibiting MPK6 in *mpk3;6* null mutants [32](S6A–S6C Fig). It is possible that some MPK3/6 activity leakage was sufficient to maintain MASS localization or other kinases may mediate MASS phosphorylation. Taken together, the combined data of *in vitro* phosphorylation and *in vivo* protein localization suggested that MAPK-mediated phosphorylation may contribute to the MASS proteins to localize robustly at the plasma membrane, though it is still possible that the manipulation of the phosphorylation sites may alter protein conformation thus protein-protein interaction for localization and function. The failure of D versions (YFP-MASS1$^{S105D}$ and YFP-MASS2.2$^{S107D}$) to localize robustly at the plasma membrane was not anticipated but suggested that dephosphorylation might be equally important for these proteins to correctly localize.

To further assess the impact of protein phosphorylation on their biological functions, we first overexpressed YFP-MASS2.2, MASS2.2$^{S107A}$, and MASS2.2$^{S107D}$ in the stomatal lineage cells by using the *BASL* promoter. Our results show that MASS2.2 did not seem to function as effectively as MASS2g in triggering stomatal clusters and neither of the mislocalized phospho-variants MASS2.2$^{S107A}$ and MASS2.2$^{S107D}$ produced significant stomatal phenotypes (Fig 5E). Considering the functional location of MASS at the plasma membrane, we modified MASS2.2 and the other two variants with the myristoylation site. When tethered to the plasma membrane, myr-GFP-MASS2.2 induced stronger stomatal clustering. However, neither myr-GFP-MASS2.2$^{S107A}$ nor myr-GFP-MASS2.2$^{S107D}$ could function at a comparable level in generating stomatal clusters (Fig 5F). Thus, we suspect that these putative MAPK phosphorylation sites need to stay open for both phosphorylation and dephosphorylation, so that MASS may achieve their function at the plasma membrane in stomatal development. Compared to the previous discoveries by Sörensson et al. that phosphomimick MASS1 trigged more stomatal production, our findings suggested that phosphorylation of MASS2 is critical for protein localization and, more interestingly, the reversible phosphorylation-dephosphorylation might be equally, if not more, important for MASS2 function.

## MASS interacts with the MAPKK Kinase YDA

The *mass* triple mutants occasionally showed a cotyledon-fusion phenotype (S6D Fig), to a certain extent resembling that of a plant expressing the constitutively active MAPKKK YDA (YDA$^{CA}$) [5]. In addition, similar to the MASS proteins, YDA is also a peripheral membrane protein in plant cells [28]. To test whether the *MASS* genes are functionally connected to YDA, we first examined the physical interaction between MASS2 with YDA. Indeed, positive protein-protein interactions were detected between YDA and MASS2 based on yeast two-hybrid and *in vitro* pull-down assays (Fig 6A and 6B). To test their interaction in plant cells, we assayed the kinase-inactive version of YDA (YDA$^{KI}$ with one point mutation K429R, [7]) because overexpression of the catalytically active enzyme often causes cell death in tobacco [33]. In the bimolecular fluorescence complementation (BiFC) assay in tobacco epidermal cells, the recovered split YFP signals supported that all three MASS proteins may physically interact with YDA$^{KI}$ at the plasma membrane (Fig 6C and S7A Fig). The interaction between YDA and MASS was further confirmed by co-immunoprecipitation (IP) analysis in tobacco leaf cells and by the biolayer interferometry (BLI) assay *in vitro* (Fig 6D and S7B Fig). In Arabidopsis stomatal lineage cells, we co-expressed mCherry-MASS2 and YDA$^{KI}$-YFP (driven by the *SPCH* promoter) and the two proteins co-existed at the plasma membrane. Furthermore, Z-projected confocal images showed that MASS2 forms cortical punctate that overlap with some YDA$^{KI}$-accumulating dots at the plasma membrane (Fig 7A), though the properties of which have not been characterized yet. Taken together, our data suggested that the MASS proteins might function through their physical interaction with YDA at the plasma membrane.

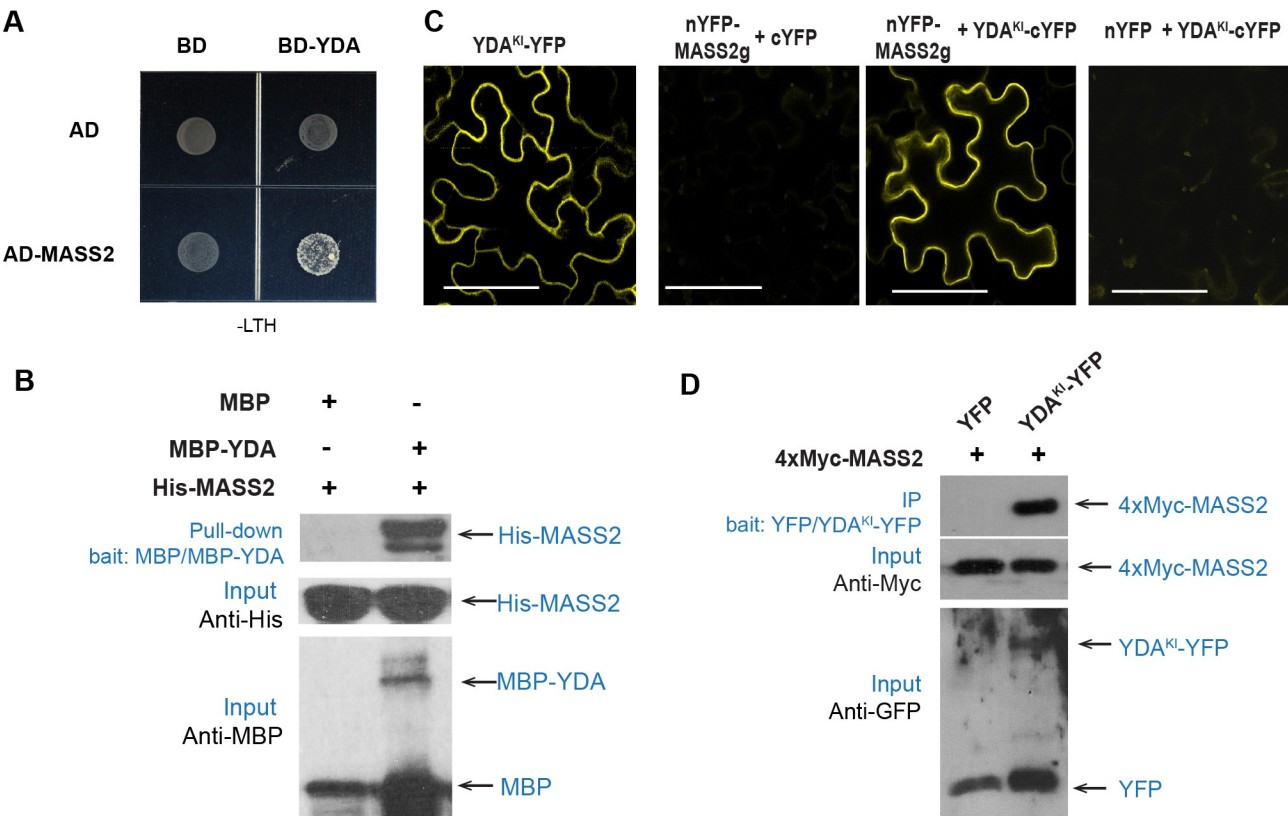

**Fig 6. MASS2 interacts with YDA.** (A) Yeast two-hybrid assay for MASS2.2 interaction with YDA. The BD and AD empty vectors were used as negative controls. Interaction tests were shown on the medium supplemented with -Leu-Trp-His. (B) *In vitro* pull-down assays using recombinant proteins, MBP-YDA and His-MASS2.3. MBP alone was used as negative control. Immunoblots were visualized by anti-His and anti-MBP. (C) BiFC assays to test the interaction between YDA$^{KI}$ and MASS2 in tobacco leaf epidermis. The expression of half YFPs (YFP$^N$ and YFP$^C$) were used as negative controls. Scale bars represent 50 μm. (D) Co-IP assay to test the interaction between YDA$^{KI}$ and MASS2. 35S::4xMyc-MASS2g was transiently co-expressed with 35S::YDA$^{KI}$-YFP or 35S::YFP in tobacco leaves.

Based on the phenotypes shown in the loss-of-function and overexpression plants, we hypothesized that the MASS family might promote stomatal production through suppressing the YDA MPK3/6 signal pathway, possibly via directly interacting with YDA. To test this hypothesis, we overexpressed CFP-MASS2g in plants expressing YDA$^{CA}$-YFP (constitutively active YDA driven by the stomatal lineage-specific *SPCH* promoter). While YDA$^{CA}$ suppresses stomatal differentiation (Fig 7B and [5]), interestingly, the introgression of CFP-MASS2 suppressed the YDA$^{CA}$-induced phenotype by restoring the formation of stomatal lineage cells (Fig 7B). Biochemically, by using the p42/44 MAPK antibody that detects activated MPK3 and 6 in Arabidopsis, we found that activated levels of MPK3/6 were elevated in the loss-of-function mutants but lowered in the *MASS2* overexpression plants (S7C Fig). However, incubation of the MASS2 protein with YDA$^{CA}$ did not seem to alter YDA kinase activity *in vitro* (S7D Fig). We suspect that either MASS2 has to be properly modified to function *in vivo* or other interacting proteins that interact with MASS to participate in the regulation of YDA MAPK signaling. At the plasma membrane, the receptor-like kinases ER could also possibly interact with MASS. When MASS2 overexpression was introduced into *er* mutants, an additive stomatal phenotype was observed (S8A Fig), suggesting that the MASS function does not seem to rely on the presence of the ER receptor. The plasma membrane-localized polarity protein BASL was also examined in MASS2 overexpression plants but no discernable changes of

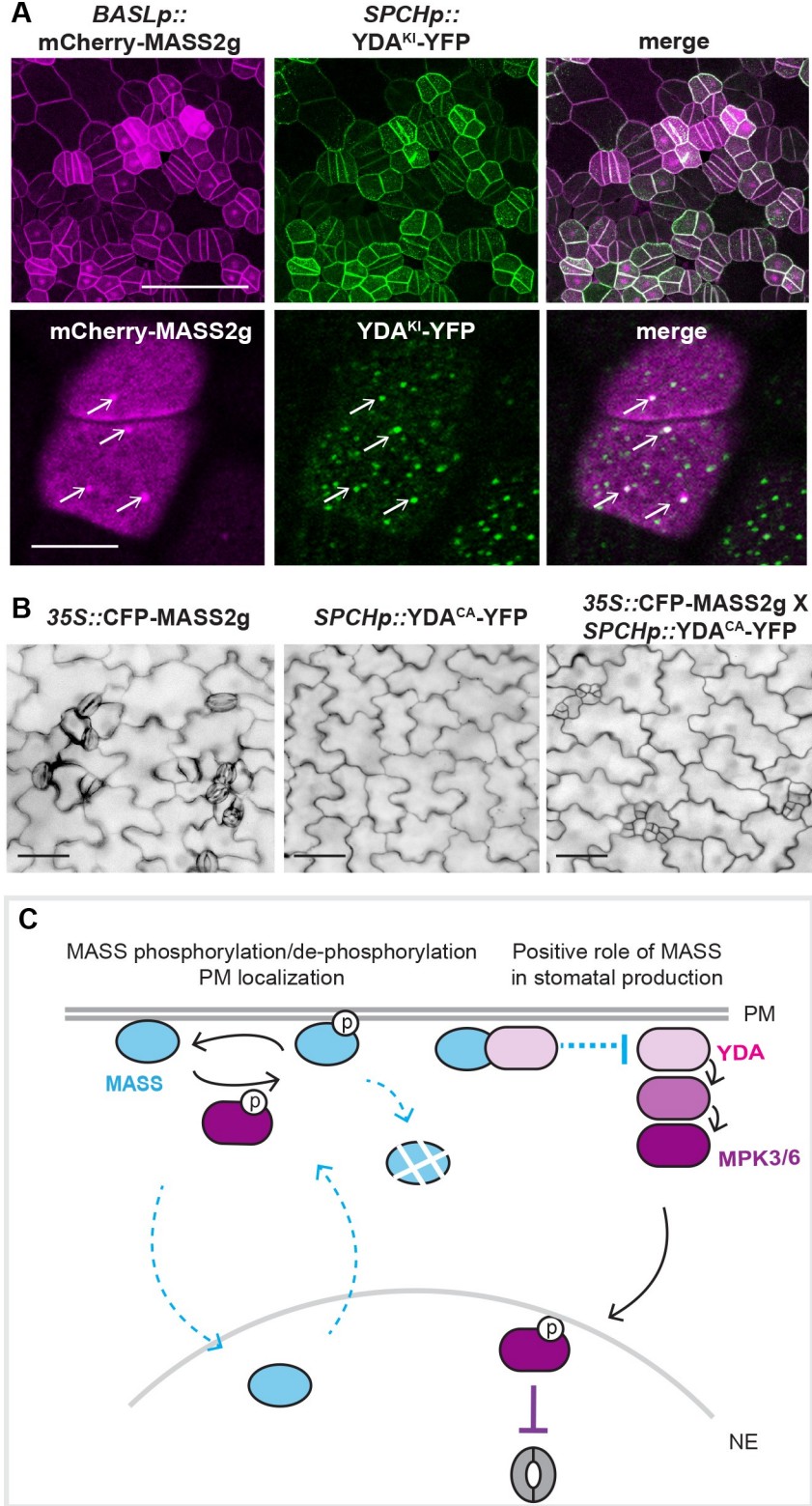

**Fig 7. Working model for MASS function in stomatal development.** (A) Confocal images showing co-localization of *BASLp*::mCherry-MASS2g with *SPCHp*::YDA[KI]-YFP in Arabidopsis. Bottom panels showing zoomed-in images for protein co-localization. White arrows indicate where co-localizations were found. Scale bars represent 50 μm in the upper panel and 10 μm in the lower panel. (B) DIC images of 5-dpg seedlings showing stomatal phenotypes generated

by *35S*::CFP-MASS2g, *SPCHp*::YDA<sup>CA</sup>-YFP, and the crossed line co-expressing both proteins. The MASS2g overexpression releases the suppression of stomatal production induced by YDA<sup>CA</sup>. Scale bars represent 50 μm. (C) A proposed working model for the MASS proteins: Protein phosphorylation and de-phosphorylation are required for MASS to robustly localize at the plasma membrane. MPK3/6-mediate protein phosphorylation may regulate both localization and function of MASS. At the plasma membrane, MASS interacts with the MAPKKK YDA, thereby directly or indirectly interfer with the MAPK signaling in stomatal development. This model does not exclude other regulators functioning with MASS to regulate stomatal development and patterning in Arabidopsis.

GFP-BASL localization were noticed (S8B Fig). Collectively, our results supported that the positive role of MASS in stomatal production might be achieved by physical interaction with YDA thus the suppression of the MAPK signaling pathway in Arabidopsis (Fig 7C). However, since MASS did not seem to suppress YDA activity directly in our *in vitro* assays (S7D Fig), we propose that MASS may function as a scaffold that recruits other regulators of the MAPK signal pathway.

## Discussion

MAPK cascades convert and amplify environmental and developmental cues into adapted intracellular responses. Their functions are particularly important for signal transduction in sessile plants that are incapable of escaping from a stressing environment. With a large number of potential kinase-substrate relationships of plant MAPKs revealed by *in vitro* and *in vivo* strategies [15, 18], most putative MAPK substrates remained functionally elusive. Plant MAPKs usually are expressed in the cytosol and/or nucleus, therefore their substrates at different subcellular localizations are thought to largely determine signal specificity and the spatio-temporal dynamics of MAPK signaling in a biological process [34]. In Arabidopsis stomatal development, SPCH and SCRM/ICE1 are nuclear transcription factors targeted by MPK3/6 for protein degradation [12–14], so that stomatal formation can be modulated by MAPK upstream signals. The polarity protein BASL in the regulation of stomatal asymmetric cell division is phosphorylated by MPK3/6 for its localization at the cell cortex where BASL functions as a MAPK scaffold protein to locally concentrate the YDA MAPK signaling to differentiate daughter cell fates [28, 35]. In this study, we established the functions of a newly identified MAPK substrate family, the MASS proteins, in the regulation of stomatal development and patterning in Arabidopsis. Phenotypic analysis of the loss-of-function mutants suggested that MASS functions to promote stomatal production and the overexpression phenotype revealed an additional role of MASS in stomatal patterning (Fig 1 and S1 Fig).

### The dual localization of the MASS proteins

The subcellular localization revealed by the fluorescent protein-tagged MASS proteins suggested that MASS1 and MASS2 are dually localized to the nucleus and at the plasma membrane, except for MASS3 that only appeared at the plasma membrane (Fig 3). But overexpression of this plasma membrane-only MASS3 induced comparable stomatal phenotypes as those generated by overexpression of MASS1, MASS2, and their myristoylated versions (Fig 1 and Fig 3), consistently suggesting the biological activities of all three MASS proteins occur at the plasma membrane.

Why doesn't MASS3 localize into the nucleus? This probably can be explained by its protein sequence, in particular at the very C-terminal end (S4C Fig). Based on our domain analyses of MASS2.3, the nuclear localization-determining fragment has been narrowed down to the C-terminal last 13 amino acids (S4C Fig) beginning with a 5 amino acid-long, basic residue-rich motif (K-R-R-S-R). This motif is fully conserved in MASS1 and MASS2 but divergent in MASS3 (S-G-G-S-T). Considering that MASS1 and MASS2 both are dually localized and

MASS3 is excluded from the nucleus, we suspect that this conserved K/R-rich motif is possibly a nuclear localization signal, which usually mediates the interaction with importins for nucleo-cytoplasmic transport [36], and that the absence of this motif in MASS3 results in the failure of the protein to localize in the nucleus.

With regards to the plasma membrane localization of MASS proteins, the highly conserved N-terminal regions aligned with the first 29 amino acids of MASS2.3 seemed to be required, though no obvious features, e.g. high hydrophobicity, lipid modification, etc., could be recognized to explain the mechanisms for the plasma membrane localization. In parallel, our work also suggested that MAPK-mediated phosphorylation of the highly conserved serine residue in the P-P-S-P motif (S4C Fig) is required for MASS to localize and function at the plasma membrane. Therefore, we propose that the factor/s determining MASS to localize to the plasma membrane may involve protein-protein and/or protein-lipid iterations with both the N-terminal 29-aa and the conserved P-P-S-P phosphorylation site.

## Possible functions of MASS at different subcellular localizations

Our findings show that, interestingly, the localization pattern and functional fashion of the MASS proteins, to some extent, mimic those of the polarity protein BASL [37]. For example, both are membrane-associated proteins that appear to be dynamically distributed between the nucleus and the plasma membrane (Fig 3A–3F) [35]. Although MASS is not polarized, both proteins function at the plasma membrane but not in the nucleus [37]. Also, both proteins are MAPK substrates and phosphorylation is important for their subcellular localization and biological function (Fig 5) [28]. In addition, both proteins appear to interact with the MAPKKK YDA at the cell cortical region. Base on the data we collected, we propose that the nuclear pool of MASS proteins, similar to that of BASL, might serve as a storage form that can be quickly targeted by MAPK signaling and redistributed to the target locations, without synthesizing new proteins, to respond to external stimuli. Meanwhile, we should not discount the possibility that MAPKs may phosphorylate MASS proteins that are in the cytoplasm or associated with the PM. Particularly, for MASS3 that is not expressed in the nucleus, the cytoplasm region can be a buffering zone. This possibility can be further tested by the deletion of the N-terminal 29 amino acids and mutating the conserved phosphorylation site.

Our complementation data using the myristoylated versions supported that the MASS proteins promote stomatal production at the plasma membrane. Then, what is the molecular mechanism for MASS to function there? We provided evidence that MASS proteins may interact with the MAPKKK YDA at the cell periphery, where MASS is hypothesized to negatively impact on the MAPK signaling cascade that suppresses the protein abundance of the key stomatal factor SPCH in production and proliferation of the lineage cells [6, 12]. However, which form (phosphorylated or dephosphorylated) of MASS may preferentially associate with YDA and how MASS proteins mechanistically suppress YDA functions, directly or indirectly, at the plasma membrane requires more in-depth investigation. There are more questions, for example, what the stability of the MASS proteins is at the plasma membrane and what controls the dynamic nuclear/cytoplasmic partition in the cells, to be addressed for better understanding the potential feedback regulation between MASS and the MAPK signaling pathway (see the model in Fig 7C).

Also, we noted that one major phenotype caused by MASS overexpression was clustered stomata, a phenotype reflecting defective cell-cell communication. In most dicot leaves, stomata are spaced out by at least one nonstomatal epidermal cells to follow the "one-cell-spacing" rule needed for efficient guard cell activity in gas exchange [38, 39]. To enforce this patterning rule, it was hypothesized that the developing guard cells release positional signals,

e.g. the peptide ligand EPF1 [40], that are perceived by the cell surface receptors, e.g. the receptor-like protein TMM [20] and the receptor-like kinase ERL1 [19, 41] with the SERK co-receptors [21], in the neighboring cells, so that the new divisions are reoriented to prevent direct stomatal contact from occurring [42]. In addition, downstream of the ligand-receptor signaling, defective YDA MAPK activities also led to the failures in enforcing the one-cell-spacing rule [5, 6]. Therefore, it is possible that the elevated expression levels of MASS at the plasma membrane may directly or indirectly alleviate the ligand-receptor signaling and/or the YDA MAPK cascade activities in the regulation of division reorientation. An expanded survey for the MASS proteins to physically interact with the individual ER and SERK family members is necessary to test this connection.

### The *MASS* genes, plant-specific and deeply conserved in early land plants

Through our sequence comparison and domain analysis, two conserved domains were recognized in the MASS protein family, including one segment at the N-terminus determining localization at the plasma membrane and another one at the C-terminus, containing the K/R-rich motif required for function and the putative MAPK phosphorylation motif P-P-S-P (S4C Fig). We aligned three MASS proteins with 40 orthologs that can be identified in land plants (embryophytes) to generate the phylogenic tree (S9 Fig). In the fern *Selaginella moellendorffii*, liverworts *Marchantia polymorpha*, as well as conifer *Picea sitchensis*, only one MASS-like (MASS-L) protein with the conserved N-terminal domain can be identified in their respective genome, suggesting their localization at the plasma membrane might be an ancient feature.

Interestingly, the typical MASS structure containing two conserved domains that appeared in the earliest flowering plant *Amborella trichopoda*, AmtMASS, and then the family members expand along with the evolution of angiosperms, e.g. 7 MASS genes in maize (S9 Fig). In addition, based on the phylogenetic assay, we found that MASS3 is more similar to ancient AmtMASS. We suspect that the full-length MASS might be co-opted from an ancient form (MASS-L) with an additional C-terminal MAPK site and the functional domain to act as a MAPK substrate. Thus, these combined features of the MASS proteins enable them as MAPK-responding regulators for plants to adapt to the developmental and environmental changes.

## Materials and methods

### Plant materials, mutants and transgenic lines

The *Arabidopsis thaliana* ecotype Columbia (Col-0) was used as the wild-type. In general, *Arabidopsis* and tobacco *Nicotiana benthamiana* plants were grown at 22°C in long days (16 h light/8 h dark). The T-DNA insertional lines *mass1* (GABI_902G09), *mass2* (SALK_061905), *mass3* (SALK_039099) were obtained from Arabidopsis Biological Resource Center (ABRC). The *GFP-BASL* marker line was described previously [37].

### Plasmid construction and plant transformation

In general, the LR Clonase II (Invitrogen)-based gateway cloning technology was used for vector construction. To generate point mutations, the plasmid pENTR/D-TOPO carrying the *MASS* genomic or coding regions were used as template and specific site mutations were introduced through a QuickChange II XL Site-Directed Mutagenesis Kit (Agilent). The entry clones were then recombined into pMDC43 (the original *35S* promoter was replaced by *BASL* promoter) and pH35CG to make *BASLp*::GFP/YFP/mCherry-MASS and *35S*::CFP-MASS, respectively. Then, the *BASL* promoter were replaced by the *MASS* promoter through *PmeI* and

*KpnI* sites to generate *MASS1p*::YFP-MASS1 and *MASS2p*::GFP-MASS2g. The pENTR/ D-TOPO carrying *MASS* promoters were recombined into pBGYN to make *MASSp*::nucYFP.

To create CRISPR/Cas9-mediated mutagenesis in Arabidopsis, we adopted the system described in [29]. By following the instructions, the oligos MASS1-CRI-F and MASS1-CRI-R were phosphorylated by T4 PNK (NEB) and annealed in a thermocycler, followed by ligation into the *BbsI* site of pAtU6-sgRNA-pAtUBQ-Cas9. Then, the chimeric U6-MASS1-Cas9 cassette was cloned into pCambia 2300 through *HindIII* and *EcoRI* sites to obtain 2300/crispr_ mass1. U6-MASS3 was amplified by PCR and inserted into 2300/crispr_mass1 through *KpnI* and *EcoRI* sites to generate 2300/crispr_mass1;3. Finally, by using the same strategies, U6-MASS2 were inserted at *EcoRI* site to create the construct 2300/crispr_mass1;3;2 to knockout the three members in the family. The *crispr* resistant MASS2 version (rMASS2) was generated through two rounds of PCR to introduce synonymous mutations in the Cas9-gRNA targeting site. Primers were listed in S1 Table.

Plasmids were transformed into *Agrobacterium tumefaciens* GV3101, which delivers the desired DNA pieces into Arabidopsis or tobacco cells. *Arabidopsis* plants were transformed with the standard floral dipping method [43, 44] and transgenic seeds were subjected to antibiotic selection. Tobacco cells were infiltrated by the method described in [45]. *A. tumefaciens* cells harboring *35S*::*CFP-MASS1/2/3* were infiltrated into *N. benthamiana* leaves and after 3 days, the leaf epidermal cells were observed under confocal microscope Leica SP5.

## Plant cell imaging and image processing

Confocal images of plant cells expressing fluorescence-tagged proteins were taken by a Leica SP5 confocal microscope. 3-dpg (day-post-germination) adaxial cotyledons of Arabidopsis were captured. Cell peripheries were visualized with propidium iodide (PI, Invitrogen). Fluorescent proteins were excited at 488 nm (GFP), 514 nm (YFP) and 594nm (PI). Emissions were collected at 500–528 nm (GFP), 520-540nm (YFP), and 620–640 nm (PI). The confocal images were adjusted using either Adobe Photoshop CS5.1 or ImageJ (Fiji). The fluorescence intensity was measured by ImageJ (Fiji) and the pixel values were export into Excel to generate the histogram graphs.

## Quantitative and statistical analysis of stomatal phenotypes in Arabidopsis

The adaxial cotyledons from 5-dpg seedlings were stained with PI and imaged were captured using a Carl Zeiss Axio Scope A1 fluorescence microscope equipped with a ProgRes MF CCD camera (Jenoptik). Stomata index (SI) was calculated as the stomata number versus the total number of epidermal cells. Clustered stomata index was calculated as the percentage of the number of clustered stomata over the total number of stomata. Stomatal clusters in Fig 1C were counted on the adaxial surface of the 10-dpg cotyledons.

## Real-time PCR

Total RNAs were extracted from 3-dpg seedlings using an RNeasy Plant Mini Kit (Qiagen). The first-strand cDNAs were synthesized by the SuperScrip First-Strand Synthesis System (Invitrogen) with 2μg of total RNAs as template in a total volume of 20 μl. The fragments of interest were amplified by sequence-specific primers (see S1 Table). Real-time PCR was performed with a SYBR Green Master Mix kit (Applied Biosystems) and amplification was monitored on a StepOnePlus Real-Time PCR System (Applied Biosystems). Gene expression levels were normalized to the reference gene (*ACTIN2)* expression using the ΔCT method. Data are presented as mean ± SD.

## Protein–protein interaction assay in yeast

The yeast two-hybrid assay was performed using the Matchmaker GAL4 Two-Hybrid System according to the manufacturer's manual (Clontech). MASS2 was inserted into pGADT7 and YDA was inserted into pGBKT7, respectively. Plasmids were transferred into the yeast strain AH109 (Clontech) by the LiCl-PEG method. The interactions were tested on SD/-Leu/-Trp/-His plates supplemented with 5 mM 3-amino-1,2,4,-triazole (3-AT). Three independent clones for each transformation were tested.

## Pull-down assay

The CDS fragments of MASS2 and YDA were cloned into pET28a or pMAL-c2x for *E.coli* expression of His- or MBP-tagged proteins, respectively. Constructs were introduced into BL21 (DE3) cells for recombinant proteins expression. The recombinant His-tagged MASS2 and MBP-tagged YDA were purified using Ni-NTA agarose (QIAGEN) or Amylose Resin (New England Biolabs), respectively, according to the manufacturer's protocol. For pull-down assays, 3 mg of MBP-YDA fusion protein was incubated with Amylose Resin at 4˚C for 2 h, the MBP tag was used as a negative control. The beads were cleaned with washing buffer (50 mM Tris-HCl, pH 7.5, 10 mM $MgCl_2$, 150 mM NaCl, and 1 mM DTT) for five times. Then the beads were incubated with 5 mg of His-MASS2 at 4˚C for 2 h. Wash beads five times with washing buffer. Western blot was used to detect the SDS-PAGE separation results of pulled-down mixtures in nitrocellulose membrane with anti-His antibody (Cell Signaling Technology) and anti-MBP antibody (New England Biolabs).

## Co-immunoprecipitation of interacting proteins in plants

*Agrobacterium tumefaciens* strains (GV3101) carrying the 35S::4xMyc-MASS2g and the 35S::YDA[KI]-YFP plasmids were co-infiltrated into *N. benthamiana* leaves. YFP fluorescence was detected 72 h after co-infiltration, and leaves were harvested and ground to powder in liquid nitrogen. Total proteins were extracted with extraction buffer (50 mM Tris pH 7.5, 150 mM NaCl, 10% Glycerol, 1 mM EDTA, 1 mM EGTA, 1 mM NaF, 1 mM Na3VO4, 10 mM DTT, 1 mM β-glycerol phosphate, 1 mM PMSF, 1 tablet/10 ml of Protease Inhibitor cocktail (Roche)). Samples were centrifuged at 14,000 rpm for 30 at 4˚C. Supernatant was subjected to immunoprecipitation by incubating with GFP-Trap Agarose (Chromotek) and rotating for 3 hr at 4˚C. The beads were washed four times with extraction buffer and the immunoprecipitates eluted with 5x Loading buffer by boiling 10 min. Total protein extracts (input) and immunoprecipitated proteins were separated on 10% SDS-PAGE and transferred to polyvinylidene difluoride (PVDF) membranes. Samples were subjected to western blot analysis with anti-Myc (1:1000; Cell Signaling Technology) or anti-GFP (1:1000; Santa Cruz Biotechnology) antibody.

## Bio-Layer interferometry (BLI) assay

The binding affinity of MASS2 with YDA was measured using the BLItz system (ForteBio Inc.), as previously reported [46]. The recombinant protein was purified and loaded onto Ni-NTA biosensors (ForteBio Inc.). Ni-NTA biosensors were first equilibrated in 50 mM Tris pH 8.0, 150 mM NaCl buffer for 10 min prior the measurements, then dipped in the buffer with purified YDA protein for the measurement of association and dissociation kinetics. The settings were as follows: initial base line for 30 s, loading for 120 s, base line for 30 s, association for 300 s, and dissociation for 500 s. The kinetic parameters Ka (association rate constant), Kd (dissociation rate constant) and the binding affinity (KD = Kd/Ka) were calculated with the help of data analysis software (BLItZ Pro). All the experiments were performed at room temperature.

Gene accession numbers in the study are *MASS1* (At1g80180), *MASS2* (At1g15400), *MASS3* (At5g20100), *YDA* (AT1G63700).

## Supporting information

**S1 Fig. MASS overexpression phenotype and expression pattern.** (A) Stomatal phenotype of MASS overexpression lines. Confocal images of 7-dpg adaxial side of the cotyledon epidermis in WT (Col) and CFP-MASS1, CFP-MASS2g, and CFP-MASS3 seedlings, all driven by *35S* promoter. Brackets indicate stomatal clusters. Scale bars represent 50 μm. (B) Confocal images to show transcriptional activities of the *MASS1/2/3* promoters displayed by the expression of nuclear YFP (nucYFP, green). Cell outlines were stained with Propidium Iodide (PI). The inset showing more detailed expression pattern. Scale bar represents 50 μm in (A) and 20 μm in (B).
(TIF)

**S2 Fig. Genetic characterization of the *mass* mutants.** (A) Diagram of the gene structure and splicing variants of the *MASS* genes. (B) Genotyping results for the *crispr* mutants. DNA sequence alignments showing the edited DNA sequences of the three *MASS* gene editing in *mass1;2;3-crispr* #5 and #7 lines. The PAM sequences were outlined with blue boxes, sgRNAs were underlined with red. (C) Genotyping PCR showing a long deletion in *MASS1* in a T3 *mass1;2;3-crispr* #7 mutant plant. (D) Quantification of SI in 5-dpg adaxial cotyledons of *mass1;2;3-crispr* #5 and #7 mutant. ** significantly different between the two samples being compared (bars). Student's *t*-test, **P < 0.001.
(TIF)

**S3 Fig. MASS1 functions at plasma membrane in stomatal lineages.** (A) *MASS1pro*::YFP-MASS1 (B) *MASS2pro*::GFP-MASS2 in T-DNA triple mutants at 3-dpg. Note, strong signals of YFP-MASS1 in the mesophyll cell layer (A), whilst strong YFP-MASS2 in the epidermis (B). (C, D) Confocal images showing stomatal phenotype in plasma membrane-localized (C) and nuclear-localized (D) GFP-MASS1 seedlings, both driven by the *BASL* promoter. Green: GFP signals, magenta: cell outlines stained with PI. Left panels show protein localization, right panels show the overlay of green and magenta. White brackets indicate stomatal clusters and abnormal cell divisions. Scale bar represents 50 μm.
(TIF)

**S4 Fig. Subdomain analysis of MASS proteins.** (A) Amino acid alignment of the MASS2 splicing variants. (B) Confocal images showing the detailed localization of GFP-tagged MASS2g, MASS2.1, MASS2.2, and MASS2.3 (green), all driven by the *BASL* promoter. Scale bar represents 10 μm. (C) Amino acid alignment of MASS1, MASS2.3, and MASS3 and the identified subdomains required for specific functions. The deleted amino acids to make GFP-MASS2.3 truncations were outlined with designated colors. (D) Confocal images of 3-dpg adaxial side of the cotyledon epidermis showing localization and stomatal phenotype of truncated MASS2.3 proteins. Cell outlines were stained with PI. Scale bar represents 20 μm.
(TIF)

**S5 Fig. Subcellular localization of MASS phospho-variants.** (A) Confocal images of YFP-MASS1 and phospho-variants shown as single optical section (s) *vs.* the z-projections (z). (B) Confocal images of YFP-MASS2.2 and phospho-variants in single optical section (s). Scale bar represents 50 μm.
(TIF)

**S6 Fig. Subcellular localization of MASS in MPK3/6-deficient background.** (A) Confocal images to show *35S*::CFP-MASS2g (red) co-expression with overexpression of the dominant negative (kinase inactive) MPK6 (*BASLp*::MPK6AEF-mRFP, green). Scale bar represents 20 μm. (B) Confocal images of *BASLp*::mCherry-MASS2 (red) co-expressed with *35S*:: MPK6AEF-YFP (green). Scale bar represents 50 μm. (C) Confocal images of GFP-MASS in chemically inducible MPK6 inhibition in *mpk3;6* null background. Scale bar represents 50 μm. (D) 5-day-old seedlings of WT, *mass1;2;3* and YDA[CA].
(TIF)

**S7 Fig. Interaction between YDA and MASS.** (A) Confocal images to show BiFC interaction tests between YDA[KI] and MASS1/3 in tobacco leaf epidermis. The expression of half YFP (cYFP) was used as negative control. Scale bar represents 50 μm. (B) BLI tests to show the interaction between YDA and MASS2. The BASL-YDA interaction was used as positive control, while MPK6-YDA as negative control. (C) Western blot to test activated MPK3/6 levels in *mass* mutants and overexpression plants. (D) *In vitro* YDA[CA] autophosporylation levels by *in vitro* kinase assay. Increasing amount of MASS2 was added to test whether it affects YDA[CA] autophosporylation activity. MKK5[KI] as a positive control, in which elevated levels of YDA[CA] trigger elevated phosphorylation of MKK5.
(TIF)

**S8 Fig. Genetic relationship between MASS proteins and other stomatal regulators.** (A) Genetic test between *er105* and a GFP-MASS2g overexpression line, driven by *BASL* promoter. Cell outlines were stained with PI (red). (B) Confocal images showing localization of Venus-BASL (green) and mCherry-MASS2g (red), both driven by the *BASL* promoter. Scale bar represents 50 μm.
(TIF)

**S9 Fig. Phylogenetic tree of the MASS family.** Protein sequences of the three Arabidopsis MASS proteins were compared with those of 40 orthologs retrieved by GenBank blasting representing embryophyta (lycophyte, gymnosperm and angiosperms). Phylogenetic tree was constructed by the program MEGA6 [47] using the neighbor-joining method. The reliability of the phylogenetic tree was evaluated by bootstrapping of 1000 replications.
(TIF)

**S1 Table. Primers used in this study.**
(PDF)

**S2 Table. Raw data for quantification.**
(XLSX)

## Acknowledgments

We appreciate helpful advice and discussion with Dr. Huiling Xue (Shenyang Agricultural University) on the phylogenic analysis of the MASS protein family. We thank Wenrui Cui (Rutgers University) for the help with data analysis. We thank the ABRC stock center for providing the T-DNA insertional lines.

## Author Contributions

**Conceptualization:** Xueyi Xue, Juan Dong.

**Data curation:** Xueyi Xue, Juan Dong.

**Formal analysis:** Xueyi Xue, Juan Dong.

**Funding acquisition:** Juan Dong.

**Investigation:** Xueyi Xue, Chao Bian, Xiaoyu Guo, Juan Dong.

**Methodology:** Xueyi Xue, Xiaoyu Guo, Rong Di, Juan Dong.

**Supervision:** Juan Dong.

**Validation:** Chao Bian.

**Writing – original draft:** Xueyi Xue, Chao Bian, Juan Dong.

**Writing – review & editing:** Xueyi Xue, Juan Dong.

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
