## [Decision Letter · Decision Letter 0]

30 Sep 2019

Dear Dr Dong,

Thank you very much for submitting your Research Article entitled 'The MAPK substrate MASS proteins regulate stomatal development in Arabidopsis' to PLOS Genetics. Your manuscript was fully evaluated at the editorial level and by independent peer reviewers. The reviewers appreciated the attention to an important problem, but raised some substantial concerns about the current manuscript. Based on the reviews, we will not be able to accept this version of the manuscript, but we would be willing to review again a much-revised version. We cannot, of course, promise publication at that time.

There are a few major issues that need to be addressed in the next version of the manuscript. First, please discuss your findings in the context of previously published literatures (e.g. Sorensson et al., Biochem J., 2012). Second, as YDA-MASS interaction is a key conclusion in this manuscript, we highly recommend that you perform Co-IP experiments to test their interaction in Arabidopsis. Third, analysis of MASS proteins driven by their own promoters is highly recommended in some key experiments. Fourth, some Z-projection data should be included in confocal analysis particularly in the cases where you exclude protein localization in some organelles. In addition, it is important to understand the effects of the YDA-MASS interaction pertaining to the questions like whether this interaction affects subcellular localization of YDA or MASS proteins, and how this interaction suppresses the inhibitory function of YDA in stomatal development.

If you decide to revise the manuscript for further consideration at PLOS Genetics, please aim to resubmit within the next 90 days, unless it will take extra time to address the concerns of the reviewers, in which case we would appreciate an expected resubmission date by email to plosgenetics@plos.org.

[LINK]

We are sorry that we cannot be more positive about your manuscript at this stage. Please do not hesitate to contact us if you have any concerns or questions.

Yours sincerely,

Hao Yu

Associate Editor

PLOS Genetics

Gregory P. Copenhaver

Editor-in-Chief

PLOS Genetics

Reviewer's Responses to Questions

**Comments to the Authors:**

Reviewer #1: Xue et al. reports a family of MAP Kinase substrate proteins, MASS proteins, that positively regulate stomatal development. The most striking finding is plasma membrane (PM) -nucleus localization dynamics of MASS and the demonstration that PM-localized MASS confers functions. All of these remind of the subcellular dynamics and MAPK regulation of BASL, which have been extensively studied by the authors’ group, although the regulatory relationship between MASS proteins and BASL are not addressed in this manuscript.

The microscopy images presented are very beautiful, and the manuscript adds very interesting family of proteins to plant specific cellular polarity modules. Having said that, there are several critical issues that need to be addressed in order to improve the manuscript.

1) Sorensson et al. report: First of all, the authors should properly acknowledge the previous work done by Sorensson et al. (Biochem J. 2012), who reported the identification of At1g80180 MASS protein (and its related protein At1g15400, now named MASS2) as in vitro and in planta MPK3/6 substrates that positively regulates stomatal development. Sorensson et al. have done quite in-depth experiments beyond this Xue et al. manuscript mentions. For example, Sorensson et al. have identified the MPK3/6 phosphosites of MASS1, performed site-directed mutagenesis to show site S105 is the phosphosite; the in vivo phosphorylation of MASS2 was reported by Scott Peck’s group; 2004 Plant Cell, which makes Xue’s phosphorylation experiments (Fig. 5A) rather confirmatory.

More importantly, Sorensson et al. performed overexpression study of MASS protein, and reported that overexpressed phosphomimic version (S105D) confers stomatal clustering and increased stomatal index in Arabidopsis. This further diminishes Xie et al.’s Result section (line 122- 1st paragraph; Figure 1), making it largely confirmatory, and the authors’ statement “we noted that one striking phenotype caused by MASS overexpression was clustered stomata” (lines 407-408) highly misleading, as it has been reported previously.

Third, Sorensson et al. has reported the uniform promoter activity of MASS in developing cotyledons using MASSpro::GUS construct. The MASSpro::nucYFP expression analysis (Fig. S1) does not provide much more insight--- What new here is that Xue et al. also characterized the promoter activity of MASS2 and MASS3.

Thus, the major conclusion of MASS proteins as positive regulator of stomatal development and MPK3/6 substrates have been reported. Here, Xue et al. have done more careful analysis using sophisticated confocal microscopy and beautiful images—however, the manuscript needs to be re-written and re-organized to properly and fairly acknowledge the previous findings.

2) Lack of analysis using endogenously expressed MASS proteins: Here all experiments of subcellular localization and interaction studies are done using ectopic promoters in the stomatal cell lineage promoters (e.g. 35S promoter, BASL promoter, SPCH promoter—Data presented in Figs. 1,3, 4,5,6,7). While use of ectopic promoters will provide “cleaner” results to interpret data-thus could be a powerful tool, however, caution should be made as it is after all ectopic expression.

I don’t see any YFP signal in Fig. 2C when the endogenous promoter was used (MASS1/2/3pro::YFP-MASS1/2/3). Is the endogenous expression that low?

Do the BASLpro:MASS1/2/3 complement the modest low stomatal index phenotype of mass1/2/3? Or does it still constitutively-active and causes excessive stomatal lineage divisions? If the latter, I would be very cautious in interpreting the subcellular localization.

Based on Fig. S1B, the promoter activity of MASS1-3 are much higher in pavement cells (and they are not expressed in stomatal precursor cells and stomatal guard cells). Where do MASS proteins localize in the pavement cells?

3) YDA-MASS interactions: The novelty (and exciting findings) of this manuscript lies in subcellular localization dynamics and precise mapping of MASS in YDA-MAPK cascade. The YDA-MASS association are tested in in vitro pull down, BiFC and Y2H (Fig. 6A-C), all highly sensitive yet occasionally face false positives, and lacking in vivo subcellular contexts. The co-localiation of YDA and MASS in plasma membrane speckle is interesting, however, again, ectopic promoters (SPCH promoter and BASL promoter) are used for expressing MASS and YDA. Overexpression of fluorescent tagged proteins are known to often form aggregates, and I think it is important for the authors to perform Co-IP experiments of YDA-MASS interactions using endogenously expressed (i.e. epitope-tagged proteins driven by the endogenous promoter) YDA and MASS.

3) Genetic interactions of MASS with YDA-MAPK cascade: Again, the genetic studies are done with ectopic expression of MASS vs ectopic expression of YDA (Fig. 7), which could be misleading. The authors need to present genetic analysis using loss-of-function mutants, which properly places the components in the genetic pathway. Because the phenotype of mass1/2/3 triple loss-of-function mutants is so subtle/weak, it may be masked by strong yda loss-of-function phenotype. However, the authors could test mass1/2/3 interactions with mpk3 and mpk6. Since mpk3 and mpk6 single mutants have no stomatal phenotype (or slightly elevated stomatal index, as reported by Putarjunan et al. 2019), the mass1/2/3 could counteract with the modest elevated stomatal index phenotype of mpk3 and mpk6.

4) Subcellular localization of MASS proteins in YDA-MAPK mutants: The authors mention that neigher phosphomimic or phosphonull substitutions of MASS proteins properly localize to PM (Fig. 5; lines 266-268). What are the subcellular localizations of YFP-MASS1/2/3 in mpk3/6 double and YDA mutants? This need to be characterized to confirm that YDA-MPK3/6 are indeed required for proper PM localization of MASS.

5) Positive role of MAPK: In the Intro (and at some extent in Discussion), the authors rationalize their study mentioning that MAPK pathway components that positively regulate stomatal development is not well studied (lines 77-79). The Lampard et al. 2009 and 2014 Plant Cell papers show that the YDA-MKK7/9-MPK3/6 module promotes stomatal development when expressed in the later stomatal precursor cells (GMC-to young guard cells, driven by the FAMA promoter). None of the expression studies of Xue et al. manuscript are done in the later stage of stomatal precursors. As far as I can see from Fig. S1, the MASS1/2/3 promoters are not even active in the GMC-young GCs. How could the authors imply the role of MASS proteins in the known positive pathway?

6) Intro-ICE1: Recent study identifies ICE1 as a scaffold to recruit MPK3/6 to nucleus, to bridge to SPCH, which does not physically associate with MPK3/6 (Putarjunan et al., 2019). Thus, the Introduction mentioning that “ICE1 is also phosphorylated by MPK3/6 (line 86-)” (and also related sentence in Discussion) is not properly reflecting the current knowledge.

Minor: Fig. 5A “CCB” should be “CBB”?

Reviewer #2: The authors present a considerable body of work examining the role of MASS proteins in stomatal development. Along with mutant and overexpressors, the study examines localization and motif analysis as well as protein-interaction studies. The study concludes that the MASS proteins represent novel positive regulators of stomatal development that act in a localization dependent manner to inhibit the activity of the MAPKKK YDA. Overall, this body of work presents novel insight into the MAPK module that regulates stomatal development and is clearly present and discussed. There are however, components of the study that require clarification or expansion to fully justify the author’s conclusions.

Minor comments

SI graphs would be better presented as violin plots or box and whiskers to show the distribution of the data.

qRT-PCR expression analysis graphs should also show the data points. Both of these are relatively common requests with regards presentation.

P8 Line 145 ‘….mutant produced lowered density of stomatal guard cells.’ Here the authors are referring to SI data and so this should reflect that this is a proportion and not density.

Figure 2C/F. The authors refer to the complementation lines in the insertion mutant but the SI data for these lines is missing from 2F.

The various fusion proteins and the resulting microscopy is given as proof of localization. For clarity, it would be good to see some blots probed with a GFP-antibody to demonstrate that these are fusions and that there is not significant free-GFP contributing to the signal.

P15 line 295. Can you really not show the data here – if it resembles the CA-YDA then it would be good to see.

Major comments

Confocal imaging is vital here for demonstrating the localization but there is little detail in the methods regarding the image capture and post-capture processing. To demonstrate fully the different localisations seen, some Z-projections are required in supplemental data. This is particular the case when the argument is that the protein is only at the membrane or the nucleus – a single plane is not sufficient to demonstrate this.

Interactions with YDA. Personally, for in planta interaction, a Co-Ip would be stronger proof than the in vitro method (not in planta), BiFC or the co-localisation. Certainly, these two latter results could potentially be explained by interactions with MPK6 and this being in close proximity to YDA.

The authors utilize the correct methodology to examine the role of phosphorylation (generating phosphomimics and mutants). However, the results are confusing. I’d like to add that the authors have been clear in reporting this and it’s good to see these results reported. However, I think that the resulting interpretation is only one possible explanation. If the argument is that phosphorylation is important for targeting then yes, you would hope to see differential localization. The alternative explanation is that the amino acid changes have both altered function independent of any role of phosphorylation. What would perhaps be useful in resolving this would be an analysis of the CFP-MASS2 localisation when crossed with the constitutively active version of YDA (7a). Here, you would at least expect enhanced MPK6 activity and MASS phosphorylation – so do you see an overall change in membrane versus nuclear localization?

On a similar front. If the expectation is that these MASS proteins inhibit YDA and therefore activation of the MAPK pathway, you might expect to see higher basal MPK6 activity in the mass triple mutant and lower MPK6 activity in the OE lines – this can be tested with phosphor specific antibodies. This could also be correlated with gene expression analysis of SPCH and some of its targets.

In their discussion, the differential localization of BASL is discussed and the similarities drawn with the localization of the MASS proteins. Linking to above, a question that arises is therefore, where is MPK6 phosophorylation these MASS proteins? In the nucleus or is it the cytoplasm or when they associate with the PM? The author’s discussion suggest they favour a nuclear phosphorylation event. I’m not suggesting that they determine this though I believe that they have the tools to do so. However, they could therefore slightly expand their discussion here to discuss the relevance to signalling of this model – they suggest it is a pool for responding to signals but the implications are that this would be a potential feedback mechanism to bring the MAPK signal back to normal. The question would then be what the stability is of the MASS proteins at the PM and whether they are dephosphorylated and then re-targetted or just broken down. Basically, their model could be expanded on.

Reviewer #3: In their manuscript, Xue et al., identified a novel family of small proteins that are involved in stomatal development. A member of this 3-protein family named MASS was previously found to be a substrate of MPK3/6. Through overexpression and mutant analyses, the authors first demonstrated that the MASS proteins act to promote stomatal production. Fluorescent reporters showed that the MASS proteins are localized either to both the plasma membrane and the nucleus or exclusively to the plasma membrane, and that they exert their function on the plasma membrane. Deletion and amino acid substitution analyses further identified that specific protein domains and the MPK6-targeted phosphorylation site on MASS2 are important for its subcellular localization and function. Finally, the authors found that MASS proteins physically interact with the triple MAP kinase YDA and appear to suppress the inhibitory function of YDA in stomatal development.

The YDA-mediated MAP kinase pathway is a key module in regulating stomatal development and patterning, as well as in other developmental processes. Thus, it is of great interest to understand how YDA is regulated. The identification of the MASS proteins as new stomatal regulators acting through YDA provides novel insights into the regulation of this critical signalling pathway. The functional dissection of the domains and the phosphorylation site also yielded valuable information about these proteins. In addition, the quality of data and the writing are generally good. To further strengthen manuscript, here are a few points that I would like the authors to address:

1. The authors mentioned that the overexpression of MASS proteins produced “overproliferated early stomatal lineage… cells” (Line 131) but did not show any quantitative data. Since these “early lineage cells” are highly relevant to YDA activity, I think the authors should quantify them in young cotyledons/leaves in the overexpression and the triple mutants.

2. The stomatal index data for mass1;2;3-crispr#5 and rescued lines were mentioned but are missing (Line 164 and Line 172).

3. For the BiFC (Fig. 6C and S5A), there should be another negative control with nYFP and YDA-KI-CFP.

4. The kinase inactive YDA was previously shown to cause a dominant negative phenotype (i.e. over-proliferation of the stomatal lineage) (Lampard et al., 2009). If the authors have the relevant data, I think the authors could indicate whether the overexpression of MASS2g (Fig. 6D) can further enhance the dominant negative phenotype.

5. Since the authors are the first to characterize this protein family, it would be helpful to the community to indicate if the mutants have other obvious growth defects and if these proteins are expressed in other tissues or developmental stage.

Others:

- The AGI code for MASS3 in the text (Line128) and the figure is not correct (compared with Line 452)

- In Fig. 6B, the authors should indicate more clearly in the figure or legend that MBP and MBP-YDA were used as baits.

**Have all data underlying the figures and results presented in the manuscript been provided?**

Reviewer #1: Yes

Reviewer #2: No: The complementation data appears to have been left out of one graph and they do say data not shown elsewhere for a phenotype.

Reviewer #3: None

PLOS authors have the option to publish the peer review history of their article (what does this mean?). If published, this will include your full peer review and any attached files.

Reviewer #1: No

Reviewer #2: No

Reviewer #3: No

---

## [Decision Letter · Decision Letter 1]

6 Feb 2020

Dear Dr Dong,

Thank you very much for submitting your Research Article entitled 'The MAPK substrate MASS proteins regulate stomatal development in Arabidopsis' to PLOS Genetics. Your manuscript was fully evaluated at the editorial level and by independent peer reviewers who reviewed the last version of this manuscript. The reviewers appreciated your revisions, but identified some aspects of the manuscript that should be further improved.

We therefore ask you to modify the manuscript according to the review recommendations before we can consider your manuscript for acceptance. Your revisions should address the specific points made by Reviewer 2. Based on the results presented in the manuscript, you may consider to tone down the conclusions in the MPK3/6 regulatory role in mediating MASS activity and localisation.

[LINK]

Yours sincerely,

Hao Yu

Associate Editor

PLOS Genetics

Gregory P. Copenhaver

Editor-in-Chief

PLOS Genetics

Reviewer's Responses to Questions

**Comments to the Authors:**

Reviewer #2: The authors have provided a much improved manuscript with new data to address the concerns raised by this reviewer.

There are two major aspects to consider here. Firstly, the MPK3/6 aspect and secondly the YDA aspect. With regards YDA, the authors provide new and strong data to support their conclusion that the MASS proteins interact with YDA. Whilst it is not clear how this impacts on YDA activity based on the authors experiments (S7D), the genetics (Fig 7B) and the phospho-kinase blot (S7c) suggest that YDA-MKK-MPK activity is impacted. With regards S7C, it is customary to strip and reprobe with the anti-MPK6 antibody to show loading rather than show gel loading.

Where there is less clarity is the role of MPK3/6 phosphorylation. The authors maintain that MPK3/6 mediated phosphorylation of Ser 107 is impacting localisation and function however, the data is still inconclusive. Certainly, mutation of this residue interferes with PM-nuclear partitioning and activity but this is not the same as concluding that MPK3/6 phosphorylation of this residue regulates these two factors. This remains the main issue with the conclusions - the data is not conclusive in showing that MPK3/6 has this regulatory role and I think the authors should tone down these conclusions. I should add that the authors have been open in there analysis and reporting of results however, given this, I don't think this role for MPK3/6 is conclusive.

Reviewer #3: The authors have addressed all my concerns in their revision.

**Have all data underlying the figures and results presented in the manuscript been provided?**

Reviewer #2: None

Reviewer #3: Yes

PLOS authors have the option to publish the peer review history of their article (what does this mean?). If published, this will include your full peer review and any attached files.

Reviewer #2: No

Reviewer #3: No

---

## [Editor Report · Decision Letter 2]

4 Mar 2020

Dear Dr Dong,

We are pleased to inform you that your manuscript entitled "The MAPK substrate MASS proteins regulate stomatal development in Arabidopsis" has been editorially accepted for publication in PLOS Genetics. Congratulations!

Yours sincerely,

Hao Yu

Associate Editor

PLOS Genetics

Gregory P. Copenhaver

Editor-in-Chief

PLOS Genetics

Comments from the reviewers (if applicable):

**Data Deposition**

http://datadryad.org/submit?journalID=pgenetics&manu=PGENETICS-D-19-01431R2

Press Queries

---

## [Editor Report · Acceptance letter]

25 Mar 2020

PGENETICS-D-19-01431R2 

The MAPK substrate MASS proteins regulate stomatal development in Arabidopsis 

Dear Dr Dong, 

We are pleased to inform you that your manuscript entitled "The MAPK substrate MASS proteins regulate stomatal development in Arabidopsis" has been formally accepted for publication in PLOS Genetics! Your manuscript is now with our production department and you will be notified of the publication date in due course.

With kind regards,

Jason Norris

PLOS Genetics

On behalf of:
